# The Emotional Impact of Patient Loss on Brazilian Veterinarians

**DOI:** 10.3390/vetsci11010003

**Published:** 2023-12-19

**Authors:** Simone Moreira Bergamini, Stefania Uccheddu, Giacomo Riggio, Maria Rosa de Jesus Vilela, Chiara Mariti

**Affiliations:** 1Etoclinvet, Rua da Quitanda 19/911, Rio de Janeiro 20011-030, Brazil; smb_6@hotmail.com; 2San Marco Veterinary Clinic and Lab, Behavioral Department, Viale dell’Industria 3, 35030 Veggiano, Padova, Italy; stefania.uccheddu@sanmarcovet.it; 3Department of Veterinary Medicine, University of Perugia, Via San Costanzo 4, 06126 Perugia, Italy; giacomoriggio@gmail.com; 4Online Psychotherapy, Rio de Janeiro, Brazil; rosavilelapsi@gmail.com; 5Department of Veterinary Sciences, Università di Pisa, Viale delle Piagge 2, 56124 Pisa, Italy

**Keywords:** veterinarians, euthanasia, burnout, grief, veterinarian training

## Abstract

**Simple Summary:**

Unlike human physicians, veterinarians may follow their patients throughout their lifetime, from conception to end-of-life care. Due to the high involvement with the animal, healthcare providers (veterinarians and veterinary staff) are likely to be exposed, together with the caregivers, to the impact of negative events including death and euthanasia. The degree course of veterinary medicine should contain a course that prepares future veterinarians for bad news delivery in order to better handle these challenging situations. Along with the need to further explore this topic, veterinarians should be taught how to recognize psychological fatigue and avoid burnout, seek medical and psychotherapeutic advice, look for ways to lessen these effects and make it easier to identify workers who are most susceptible to burnout early on.

**Abstract:**

Veterinarians, unlike human physicians, could potentially care for the patient for several years, from conception to end-of-life care. Because of their close relationship with the animal, healthcare providers (for example, veterinarians and staff) are more likely to be affected by bad events and end-of-life care. The purpose of this study was to assess the emotional impact of patients’ deaths on Brazilian veterinarians; 549 Brazilian veterinarians (78.3% females) completed a 20-item online questionnaire. Females were more emotionally affected than males by having to talk to the owner about their animal’s death and more emotionally affected by the animal’s death itself. Furthermore, the emotional impact of an animal’s death was heavily influenced by the number of animals euthanized and varied greatly across veterinarians based on their age, with vets over 50 years old being less affected than vets between the ages of 31 and 40. The majority of responders (91.0%) were not trained to deal with grief during their degree. Those who had some training reported being less affected by bereavement. These findings indicate that patient death is a significant emotional concern for veterinarians. Specific education during the degree course, aimed at preparing future veterinarians to deal with death and death communication, is lacking but necessary.

## 1. Introduction

Unlike human physicians, veterinary practitioners may follow their patients throughout their lifetime. In fact, it is common for a veterinarian to be the referring doctor of a patient for several years, possibly from conception to end-of-life care. Because of this type of relationship with the animal, veterinary care providers (veterinarians, for example, and the staff) are likely to be emotionally exposed, together with the caregivers, to the impact of adverse events. The term “first and second victims” has been coined to better describe this phenomenon. While the first victim, that is, the caregiver [1], is directly affected by an adverse event (e.g., pet’s death), the second victim, represented by the veterinarian or the veterinary staff, may be traumatized by the regular involvement in ending the lives of their patients or by taking on the burden of a dying or ill pet [2,3].

In addition, veterinarians often develop an intense relationship with the client (caregiver), which may make them more emotionally involved and more empathetic to their clients’ emotions related to the suffering and the loss of the pet [3,4,5]. A very high percentage (between 88% and 99%) of pet owners see their animals as family members [6,7,8]. Rising acknowledgment of pets as family members has been associated with pet owners’ increasing expectations for the highest quality veterinary care, as well as compassionate care and respectful communication for themselves. Psychosocial interventions for the first victim have been investigated in recent years [9,10,11,12,13]. However, second victims may require the same support in the veterinary setting [14] to limit psychological fatigue and avoid burnout. Traumatic events, repeated over time, may affect coping mechanisms and opportunities for stress relief, resulting in exhaustion, distress, feelings of guilt, shame, helplessness, inadequacy, and loss of adaptive functioning and unfortunately increasing the percentage of suicide [1,14].

Although it is not the scope of the veterinary profession to treat psychological issues in pet owners, it is important for veterinarians to understand whether their clients are experiencing any kind of distress in the context of their professional relationship [13]. 

Kahler [15] reports that even giving bad news and interacting with difficult clients is a source of high stress. Euthanasia can also result in significant levels of stress for both the veterinarian and the caregiver. Despite euthanasia being a common process in veterinarians’ professional practice, research suggests that veterinarians do not feel they have been prepared well for the task during their education [16].

When it comes to the decision-making process of euthanasia, negative dimensions of bereavement (grief, guilt, anger, intrusive thoughts, and decisional regrets) are strictly linked to each other [17]. The clients consider the veterinarian as a reference figure and after the loss of the animal, they expect the veterinarian to provide support and guidance [18]. However, Dickinson et al. [19] also report that many veterinarians are not trained adequately in how to deal with pet loss and clients’ emotions, not only when it comes to euthanasia but likewise when veterinarians cope with a patient’s natural death. 

Recently, decision-making processes related to pets’ end-of-life care have sparked some interest among veterinary researchers, who generally accept that, in such situations, veterinary practitioners should adopt the role of educators and counselors to the client rather than making the decision on the client’s behalf [20]. Effective client support during and after a patient’s death is characterized by empathic communication. Veterinary care should prioritize improving patient comfort and minimizing suffering while fostering a cooperative and supportive collaboration with the caregiver client [16]. Relationship-centered care, which is defined as a partnership in which the veterinarian and the client use discussion and collaborative decision-making to incorporate the client’s perspectives and recognition of the importance of animals in the client’s life into all aspects of care, has been suggested as the ideal model for veterinarian–client relationships, just like in human medicine [17]. 

Most veterinarians realized that euthanasia could have such a big impact on them, on their staff members, on their clients, and on the quality of their relationships between veterinarians and clients [21]. When euthanasia is conducted recklessly, unconsciously, or without compassion and sensitivity, it may complicate and prolong suffering and harm and even end the client–veterinary relationship due to resentment and anger [22]. Considering the emotions and stress that accompany the veterinary profession, coping strategies have been investigated (e.g., talking with someone, contacting the owner after the death) in order to understand the best strategies that veterinarians could apply [23]. There is a relationship between compassionate fatigue, secondary traumatic stress, and burnout in all professionals who work in situations of trauma or suffering such as veterinarians [24].

The roles and responsibilities of veterinarians and their staff during the transition and enactment of end-of-life care are also extensive and multifaceted [21]. The abilities required for excellent end-of-life treatment have gone beyond what many veterinarian offices are qualified to deliver. Under any circumstances, the healthcare team must communicate with a unified voice and sense of purpose, especially when end-of-life care is involved. In end-of-life situations, each member of the veterinary healthcare team should have clearly defined caregiving and client-support roles, particularly ones that make use of individual abilities, strengths, and experience [25]. 

In this respect, the aim of this study was to evaluate the emotional impact of the loss of a patient on Brazilian veterinarians. 

## 2. Materials and Methods

A 20-item online questionnaire was prepared to be completed by Brazilian veterinarians. All questions were close-ended, so respondents were allowed to choose one (all but items 10, 11, and 15) or more answers from a given list of answer options. All items were mandatory; only item 15 had to be answered by respondents who chose “yes” to item 14.

The link to the questionnaire was circulated via email, WhatsApp, and Facebook groups for veterinarians in 2017. Before releasing the questionnaire, five people were asked to read and confirm the clarity of the questions. The original questionnaire in Portuguese and translated into English is added as supplementary material in Appendix A.

This research was approved by the Ethics Committee for Research of the Educational Center Augusto Motta in Rio de Janeiro, number 5.282.227.

### Statistical Analysis

Statistical analysis was performed using SPSS Statistics version 25 (IBM, SPSS Inc., Chicago, IL, USA). Descriptive statistics were performed for all demographic variables. In order to investigate different aspects of the veterinarians’ emotional response to the death of a patient, three main items were selected for further analysis, namely item (9) “In both cases (natural or recommended for euthanasia) how do you feel when reporting the animal’s death to the owner?”, item (13) “In relation to the animal (patient), how do you feel when you need to do euthanasia or cannot save her/him?”, and item (18) “A few days after the animal’s death, you get in touch by phone, send messages or cards to the owner showing support or solidarity with the loss of the animal?”. The three questions were chosen as they relate to different aspects of the patient’s death, respectively: (9) empathy towards the owner), (13) sorrow for the animal’s death, and (18) support to the owner.

Preliminary non-parametric statistics (i.e., Mann–Whitney and Kruskal–Wallis) was used to identify differences in the participants’ responses to these three items according to demographic variables, such as sex, age, time from graduation, specialization, times they talk about the patient’s death/euthanasia with the owner, if and whom they tried to talk to, and if they had been educated on communicating death during their university course. The responses to the question on specialization were grouped into two categories for statistical purposes, namely “general practitioners” and “specialized veterinarians”. For the same reason, the responses to the question on whether they had tried to talk to someone about their feelings, and to whom, were categorized as either “to nobody”, “to a non-professional (e.g., friend, relative, colleague), or “to a professional”. The level of statistical significance was set at <0.05 and a Bonferroni correction was applied for pairwise comparisons. For each of three selected items, a Generalized Linear Model (GLM) with a multinomial distribution and cumulative logit function was built using the demographic variables that resulted in significant differences in the previous analysis as predictors. Specifically, the variables included as predictors for the first item were gender and education on communicating death during the university course; for the second item, they were gender, age, and times they talk about the patient’s death/euthanasia with the owner; for the third item, they were gender, age, specialization, and times they talk about the patient’s death/euthanasia with the owner. The variable “time of graduation” was excluded from the models—although significantly different for items 13 and 18—because it was found to be highly correlated with age (τ = 0.729). The variable “attempts to talk to someone about feelings” was also excluded from the models as we thought it would not be conceptually correct to treat this variable as a predictor. Multicollinearity was checked before performing the regression analysis. Finally, Chi-square was used to investigate the association of being educated on communicating death during the university course with the veterinarians’ strategy when notifying the animal’s death to the owner, as well as with their emotional state when the owner gives the pet the last goodbye. 

## 3. Results

### 3.1. Demographics

In total, 549 veterinarians completed the questionnaire. The majority were women (n = 430, 78.3%), with men representing about one-fifth of the sample (n = 119, 21.7%). A large proportion of them were between 31 and 40 years old (n = 230, 41.9%), followed by 20–30 (n = 197, 35.9%), 41–50 (n = 90, 16.4%), and above 50 years old (n = 32, 5.8%). A similar number of respondents (n = 173, 31.5%) had graduated in veterinary medicine less than 5 years (n = 173, 31.5%) and between 5 and 10 years before their participation in this study (n = 172, 31.3%), followed by veterinarians who graduated between 10 and 20 years (n = 145, 26.4%) and more than 20 years before (n = 59, 10.7%). The great majority of the veterinarians worked in small animal clinics (n = 548, 93.5%). General practitioners represented approximately half of the sample (n = 275, 50.1%), whereas the most frequently reported specialties were feline (n = 48, 8.7%) and exotic animal medicine (n = 15, 2.7%). However, 25.4% of the veterinarians reported having an unspecified specialty (n = 131, 23.9%). 

### 3.2. Emotional Impact of Patient Loss and Euthanasia 

According to the results of the preliminary analysis, female veterinarians were more emotionally affected by having to euthanize a patient or by not being able to save them compared to male colleagues (U = 34,015.000, *p* < 0.001). Furthermore, the emotional impact of the animal’s death was significantly different across veterinarians according to their age (H(3) = 9.505, *p* = 0.023), with vets aged more than 50 years being less affected than vets aged between 31 and 40 (U = 77.460, *p* = 0.035), as well as to how long had passed from the time of their graduation (H(3) = 10.114, *p* = 0.018), with vets who graduated more than 20 years prior to the study being less affected than those graduated between 5 and 10 years prior (U = 69.280, *p* = 0.012). The emotional response to the patient’s death also differed depending on the times they talked about the patient’s death/euthanasia with the owner (H(3) = 12.105, *p* = 0.007), with veterinarians who talked about death/euthanasia for more than 50 animals being less affected than those who did it for less than 10 animals (U = 52.050, *p* = 0.016) and those who did it for 11 to 30 animals (U = 48.025, *p* = 0.021), as well as on their attempts to talk to someone about their feelings (H(2) = 15.089, *p* = 0.001) with veterinarians who did not try to talk to anybody being less affected than those who tried to talk to relatives, friends, colleagues (U = 40.672, *p* = 0.019), or professionals (U = 78.837, *p* > 0.001). 

Similar results were obtained with the GLM (AIC = 312.819, χ^2^_7_ = 51.410, *p* < 0.001), where female veterinarians were more likely (OR = 0.327, 95%CI = 0.220–0.484, *p* < 0.001) to feel emotionally affected by the death of a patient compared to males. Similarly, veterinarians between 31 and 40 (OR = 2.323, 95%CI = 1.127–4.789, *p* = 0.021) were more likely to feel emotionally affected by the death of a patient than those older than 50 years. Finally, those who had to talk about death/euthanasia for less than 10 patients (OR = 1.702, 95%CI = 1.083–2.674, *p* = 0.021) and for 11 to 30 patients (OR = 1.595, 95%CI = 1.052–2.419, *p* = 0.028) were more likely to feel emotionally affected by the death of an animal than veterinarians who had to do it for over 50 patients. All the GLM results are summarized in Table 1.

Overall, a great majority of the respondents (n = 430, 78.3%) reported that they had tried to talk to someone about their feelings in relation to the loss of a patient. A summary on the type of person veterinarians had tried to talk to about their feelings and the relative frequencies is reported in Table 2.

### 3.3. Emotional Response to Communicating the Patient’s Death to the Owner

According to the results of the preliminary analysis, female veterinarians were more emotionally affected by having to notify the owner of their animal’s death compared to male colleagues (U = 32,986.000, *p* < 0.001). Furthermore, the veterinarians’ emotional response to having to notify the death of an animal to the owner differed according to their attempts to talk to someone about their feelings (H(2) = 9.797, *p* = 0.007). Specifically, veterinarians who had not tried to talk to anybody about their feelings were less emotionally affected than those who had, either to a friend or a relative (U = 41.270, *p* < 0.00127), or a professional (U = 63.095, *p* < 0.012). Finally, veterinarians who did not receive specific education on how to deal with communicating the death of a patient to the owner were more emotionally affected than veterinarians who received that type of education during their university course (U = 13.653, *p* < 0.044). According to the results of the GLM (AIC = 55.486, χ^2^_2_ = 34.754, *p* < 0.001), only gender had a significant effect on the veterinarians’ emotional response to having to report a patient’s decease to the owner, with women being more likely (OR = 0.301, 95%CI = 0.192–0.471, *p* < 0.001) to feel emotionally affected compared to men. A summary of the GLM results is presented in Table 3.

Overall, veterinarians seem to apply different strategies to communicate the death of a patient to the owner. For instance, 77.8% of the respondents reported that they try to comfort the owner with friendly words, 70.1% use technical explanations about the clinical problem, 12.4% use religious arguments, 12.9% prefer not to talk too much, and 1.3% ask another person to talk to the owner. Among these, only trying to comfort the owner with friendly words was associated with having received specific education on pet death communication during the university course. Veterinarians who did not receive it were more likely to report this type of strategy (χ^2^ = 24.104; *p* = 0.014). 

Similarly, veterinarians reported different emotional responses when watching the owner giving their last goodbye to the pet (Table 4). Again, the specific type of emotional response was related to having been educated on how to communicate the patient’s death during their university course. Veterinarians who did not receive specific education on this topic were more likely to cry along with the owner (χ^2^ = 7.573; *p* = 0.006), have a trembling voice (χ^2^ = 4.620; *p* = 0.032), and feel guilty for not being able to save the animal (χ^2^ = 5.129; *p* = 0.024), and were less likely to always feel in control of their emotions in that situation (*p* = 0.004).

### 3.4. Solidarity Behaviors towards the Owner after the Animal’s Death 

According to the preliminary results, male veterinarians showed some support or solidarity to the owners after the animal died by sending cards or giving phone calls more often than female colleagues (U = 21,975.000, *p* < 0.007). The frequency of the veterinarian’s solidarity behavior towards the owner also differed according to their age (H(3) = 15.686, *p* = 0.001) and the time passed from graduation (H(3) = 11.331, *p* = 0.010). Specifically, veterinarians aged 20 to 30 years display some form of solidarity behavior more often than veterinarians older than 50 years (U = 90.175, *p* < 0.004). Similarly, veterinarians who graduated less than 5 years ago showed solidarity behaviors more often than those who graduated more than 20 years ago (U = 61.739, *p* < 0.020). Furthermore, general practitioners reported a higher frequency of solidarity behaviors compared to colleagues with a specialization (U = 34,084.500, *p* < 0.028). The frequency of solidarity behaviors was different amongst veterinarians depending on the times they had to talk about the patient’s death/euthanasia to the owner (H(3) = 8.922, *p* = 0.030), with veterinarians who had to do it for more than 50 patients showing such behaviors less frequently than those who had to do it for less than 10 of them (U = 46.463, *p* < 0.025), as well as on their attempts to talk to someone about their feelings (H(2) = 13.592, *p* = 0.001), with veterinarians who talked to professionals showing less solidarity behaviors than those who did not talk to anybody (U = −75.523, *p* < 0.001). 

The GLM (AIC = 317.163, χ^2^_8_ = 35.154, *p* < 0.001) yielded consistent results with male veterinarians being more likely than females (OR = 1.990, 95%CI = 1.286–3.079, *p* = 0.002) to express some solidarity to the owner after the animal died with follow up calls or cards. Veterinarians between 20 and 30 years old (OR = 3.677, 95%CI = 1.672–8.091, *p* = 0.001) and veterinarians between 31 and 40 (OR = 2.707, 95%CI = 1.262–5.806, *p* = 0.011) were also more likely to express some solidarity towards the owner compared to colleagues older than 50 years, and so were general practitioners compared to specialists (OR = 1.408, 95%CI = 1.000–1.983, *p* = 0.050). Finally, veterinarians who talked about the patient’s death/euthanasia with the owner for less than 10 animals (OR = 1.756, 95%CI = 1.075–2.869, *p* = 0.025) were more likely to display solidarity behaviors compared to those who had to do it for over 50 animals. All the GLM results are summarized in Table 5.

## 4. Discussion

The emotional impact of the loss of a patient on veterinarians is an innovative topic that deserves much attention due to its potential consequences. Unlike many other topics that have been largely investigated in human medicine and then replicated in veterinary medicine, this topic has received little attention in the scientific human literature [26], probably due to the low number of countries that allow physician-assisted suicide; in the Netherlands, where human euthanasia can be performed, the emotional burden of preparing and performing euthanasia or assisted suicide was commonly reported by physicians [27], similar to what was observed in our study. At first appearance, human and (companion animal) veterinary medicine appear to share difficult processes in end-of-life (EOL) decision-making. However, there are significant differences between the therapeutic alternatives offered by the two professions [28]. The ethical implications are likely more complex in human medicine, and, together with the lack of literature, do not allow for a direct comparison of results obtained here for veterinarians with a similar situation for physicians. 

The routine of veterinary practitioners includes discussions with clients concerning euthanasia and the end of the animal’s life. According to our findings, the majority of veterinarians are emotionally affected when they have to tell the client about the animal’s death, with a quite high percentage of them reporting high levels of distress. There are bad news protocols that are utilized for health professional training and can be used by experts to approach patients with serious and terminal illnesses [23]. According to this study, veterinarians who have received death communication training report feeling less emotionally distressed when telling the animal’s owner that their pet is going to be put to sleep. Unfortunately, the survey results revealed that 91.4% of professionals did not receive enough bad news training during the course. Lesnau and Santos [29] demonstrated that academic training is little concerned with future professionals’ psychological and emotional preparation, indicating a true escape from the issue of death and mourning at Brazilian veterinary medical universities. This denial can manifest as a strategy for avoiding connection with traumatic events that professionals may encounter when caring for patients and their caregivers, which can result in a variety of problems and occupational stress. Despite this, veterinarians would have preferred to have been instructed in the graduate course on how to communicate terrible news, which could have helped them because professionals who have received proper training have been found to be less impacted by patient loss.

Concerning the emotional influence of the patient’s death on the professional, a high percentage of emotions were expressed by the professionals when they watched the owner’s farewell to the dead dog. Crying with the owner (46.8%), wanting to cry (53.4%), and tearing up (44.4%) were the most common. This large proportion of responses demonstrates the veterinarian’s deep emotional bond with the patient, as well as how stressful the death of the animal patient can be for the professional. It also implies that the expert suffers when witnessing the animal’s and human family’s suffering. According to Figley [30], professionals who frequently work with animals or people who are in danger of dying may become influenced by other people’s grief and suffering and have health issues including burnout and secondary traumatic stress. The shame over the animal’s death was one of the significant emotional responses that 38.8% of professionals in this survey reported experiencing. Sleep and eating disorders are common emotional responses to stress, and Seligman [28] claims that they are linked to low self-esteem and can result in feelings of guilt for everything that goes wrong [31].

Members of the veterinary team experience depression and other psychological suffering at significant rates [32] and professionals who are female, younger, and employed in small or mixed practices are more likely to experience stress at work and contemplate suicide [33]. Among the specialties, it was identified that generalists are more sensitive to patient loss than specialists. So, it can be expected that generalists can be more emotionally affected due to a greater number of animals with serious and/or terminal conditions or violent and unexpected animal deaths that do not reach specialists. 

The amount of time since graduation is an additional aspect that influences the veterinarians’ emotional state. The death of the patient has less impact on veterinarians who have been in practice for a while (more than 20 years) than on those who have only recently graduated (less than 5 years). The professional’s expertise appears to be a significant factor in the behavior of the veterinarian. So, if, on the one hand, having graduated more recently results in a better suitable response and stronger control of the situation due to higher security and professional expertise compared to those with less experience. On the other hand, this outcome can indicate defensive behavior in a professional as a result of prolonged exposure to stress-inducing stimuli. Zeidner et al. [34] showed that both self-reported “trait” emotional intelligence and ability-based emotion management are inversely associated with compassion fatigue.

One technique to comfort the client when the animal dies is to get in touch with them. The caregivers may see how much the staff cares for and remembers them and their animals by receiving a card or follow-up link [35]. Although women are more emotionally affected by the patient’s death, they tend not to contact the owner, probably in order to prevent emotional distress. The majority of professionals who made contact with a client after a patient passed away were men in their 20s and 30s who were generalists and assisted in the deaths of fewer than 10 animals. 

The close relationship that exists between a veterinarian and animal patients is demonstrated by the fact that 77.2% of veterinarians stated they made an effort to talk to friends, family, and coworkers about how they were feeling about the patient’s death. However, the veterinarians did not value receiving help from qualified experts and even preferred chatting to strangers over receiving therapy. Like all other health workers, veterinarians experience unacknowledged grieving since they are not allowed to communicate their feelings [33]. Thus, making an effort to communicate their feelings can help them feel less emotionally distressed. This study has potential limitations. The proportion of female respondents to male participants might represent a bias; however, it is now very common to have more females than males in the veterinarian population [36]. Since data are only collected in one country, the next step should be to consider the validity and generalizability of the results. Nonetheless, given the significance of the findings, more research is necessary to enhance veterinarian welfare.

## 5. Conclusions

Euthanasia and patient death are emotionally challenging events for veterinary practitioners. Veterinary degree courses should include mandatory education to specifically prepare future veterinarians to deal with death, as well as to communicate death or other bad news to their patients’ owners. Furthermore, veterinarians should be taught to recognize symptoms of compassion fatigue or other psychological disorders that may be caused by having to deal with suffering and death on a daily basis. This may also increase their chances of seeking medical and psychotherapeutic advice at an early stage of the disorder and identify colleagues who may be in a state of greater vulnerability, such as female or young veterinarians.

## Figures and Tables

**Table 1 vetsci-11-00003-t001:** Results of the GLM for item 13 on the emotional impact of patient loss and euthanasia. * = *p* < 0.05. B: regression coefficient. Exp (B): odds ratio. ^a^ This parameter was set to zero because it is the term of comparison.

Parameter		B	Std. Error	Sig.	Exp (B)	95% Wald Confidence Interval for Exp (B)	
						Lower	Upper
Gender	Male	−1.119	0.2010	0.000 *	0.327	0.220	0.484
	Female	0 ^a^			1		
Age	20 to 30	0.650	0.3768	0.085	1.915	0.915	4.007
	31 to 40	0.843	0.3692	0.022 *	2.323	1.127	4.789
	41 to 50	0.559	0.3979	0.160	1.749	0.802	3.815
	>50	0 ^a^			1		
Times talked about patient’s death/euthanasia	<10	0.532	0.2306	0.021	1.702	1.083	2.674
	11 to 30	0.467	0.2126	0.028 *	1.595	1.052	2.419
	31 to 50	0.202	0.2561	0.431	1.224	0.741	2.021
	>50	0 ^a^			1		

**Table 2 vetsci-11-00003-t002:** Categories of people veterinarians tried to talk to about their feelings in relation to the loss of a patient.

Category	N	% of Responses	% of Cases
Relatives	271	26.5%	63.0%
Friends	274	26.8%	63.7%
Psychotherapist and/or holistic therapists	75	7.3%	17.4%
Coworkers	320	31.3%	74.4%
Unknown people	4	0.4%	0.9%
Clients	66	6.5%	15.3%
Others	13	1.3%	3.0%

**Table 3 vetsci-11-00003-t003:** Results of the GLM for item 9 on the emotional response to communicating the patient’s death to the owner. * = *p* < 0.05. B: regression coefficient. Exp (B): odds ratio. ^a^ This parameter was set to zero because it is the term of comparison.

Parameter		B	Std. Error	Sig.	Exp (B)	95% Wald Confidence Interval for Exp (B)	
						Lower	Upper
Gender	Male	−1.201	0.2287	0.000 *	0.301	0.192	0.471
	Female	0 ^a^			1		
University training on patient’s death communication	Trained	−0.533	0.3265	0.102	0.587	0.309	1.112
	Untrained	0 ^a^			1		

**Table 4 vetsci-11-00003-t004:** Responses of the veterinarians during bad news communication.

Response	Females (% of Cases)	Males (% of Cases)	Total		
			N	% of Responses	% of Cases
I cried with the owner	235 (54.7)	22 (18.5)	257	13.1	46.8
I felt my hands and my body shaking	110 (25.6)	13 (10.9)	123	6.3	22.4
I did not cry but I wanted to	225 (57.1)	68 (57.1)	293	14.9	53.4
Tearing up	204 (47.4)	40 (33.6)	244	12.4	44.4
Headache	61 (14.2)	9 (7.6)	70	3.6	12.8
Sweat	37 (8.6)	7 (5.9)	44	2.2	8.0
Tachycardia	80 (18.6)	11 (9.2)	91	4.6	16.6
Sleep disturbance	60 (14.0)	13 (10.9)	73	3.7	13.3
Agitation	46 (10.7)	6 (5.0)	52	2.6	9.5
Difficulty to talk or trembling voice	132 (30.7)	24 (20.2)	156	7.9	28.4
Guilty because I could not save the animal	178 (41.4)	35 (29.4)	213	10.8	38.8
I prefer to be far away from the owner because I get affected easily	124 (28.8)	12 (10.1)	136	6.9	24.8
I felt angry or frustrated because I could not save the animal	120 (27.9)	35 (29.4)	155	7.9	28.2
I never had the problem with it because I have control of the situation	16 (3.7)	17 (14.3)	33	1.7	6.0
Others	17 (4.0)	7 (5.9)	24	1.2	4.4

**Table 5 vetsci-11-00003-t005:** Results of the GLM for item 18 on solidarity behaviors towards the owner after the animal’s death. * = *p* < 0.05. B: regression coefficient. Exp (B): odds ratio. ^a^ This parameter was set to zero because it is the term of comparison.

Parameter		B	Std. Error	Sig.	Exp (B)	95% Wald Confidence Interval for Exp (B)	
						Lower	Upper
Gender	Male	0.688	0.2228	0.002 *	1.990	1.286	3.079
	Female	0 ^a^			1		
Age	20 to 30	1.302	0.4023	0.001 *	3.677	1.672	8.091
	31 to 40	0.996	0.3893	0.011 *	2.707	1.262	5.806
	41 to 50	0.777	0.4187	0.064	2.174	0.957	4.940
	>50	0 ^a^			1		
Specialty	Generalist	0.342	0.1747	0.050 *	1.408	1.000	1.983
	Specialist	0 ^a^			1		
Times talked about patient’s death/euthanasia	<10	0.563	0.2505	0.025 *	1.756	1.075	2.869
	11 to 30	0.090	0.2235	0.689	1.094	0.706	1.695
	31 to 50	0.203	0.2774	0.463	1.226	0.712	2.111
	>50	0 ^a^			1		

## Data Availability

Data are available on request addressed to the corresponding author.

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
