# Peer review of "The Emotional Impact of Patient Loss on Brazilian Veterinarians"

_vetsci, 2023, doi:10.3390/vetsci11010003_

Round 1
Reviewer 1 Report
Comments and Suggestions for Authors
Hi there,
An interesting article, there are a few areas that need improvement:
1. The questionnaire would be useful to add as a supplementary material.
lines 109-111: this is an incredibly leading question, this topic could have been covered by asking "do you have a follow up procedure after euthanasia has been performed, and if so please explain what it is" or by asking something along similar lines.
lines 167-171; 174-179; 188-194 - rather confusing sentences, please try to clarify and restructure.
Lines 202-211 and their corresponding write up in the discussion (lines 315-322) - this is very unclear - did vets benefit from talking to anyone (random and professional) or not? Were they less emotionally affected if they didn't speak to anyone?
lines 293-294: this sentence is very confusing please rewrite.
Otherwise an interesting article and an important topic.
Many thanks
Comments on the Quality of English Language
There are some very confusing paragraphs, a number of typos and edits that need to be made to make the article clearer for the reader to understand.
Author Response
Dear Reviewer, thanks a lot for your comments. We have carefully reviewed your comments and have done our best to respond to each one individually, especially in terms of consistency and improved comprehension.
The questionnaire would be useful to add as a supplementary material.
Dear reviewer, the original questionnaire in Portuguese and translated into English was added as supplementary material.
lines 109-111: this is an incredibly leading question, this topic could have been covered by asking "do you have a follow up procedure after euthanasia has been performed, and if so please explain what it is" or by asking something along similar lines.
Dear Reviewer, thanks a lot for this comment, that we consider a suggestion for future research aimed at exploring more in detail the veterinarian's procedure after the animal's death.
lines 167-171; 174-179; 188-194 - rather confusing sentences, please try to clarify and restructure.
The section “results” has been dramatically changed and restructured due to the comments of all reviewers, so those sentences are not present anymore in the revised copy.
Lines 202-211 and their corresponding write up in the discussion (lines 315-322) - this is very unclear - did vets benefit from talking to anyone (random and professional) or not? Were they less emotionally affected if they didn't speak to anyone?
Lines 202-211 have been revised.
Lines 315-322 have been modified as follow (see lines 617-624): The close relationship that exists between a veterinarian and patients animal is demonstrated by the fact that 77.2% of veterinarians stated they made an effort to talk to friends, family, and coworkers about how they were feeling about the patient's death. However, the veterinarian did not value receiving help from qualified experts and even preferred chatting to strangers than receiving therapy. Like all other health workers, veterinarians experience unacknowledged grieving since they are not allowed to communicate their feelings. Thus, making an effort to communicate their feelings can help them feel less emotionally distressed.
lines 293-294: this sentence is very confusing please rewrite.
Thank you for spotting it, it was a mistake related to a previous version of the manuscript.
Otherwise an interesting article and an important topic.
Many thanks
Comments on the Quality of English Language
There are some very confusing paragraphs, a number of typos and edits that need to be made to make the article clearer for the reader to understand.
We have revised the text also for language.

Reviewer 2 Report
Comments and Suggestions for Authors
The article „The emotional impact of patient loss on Brazilian veterinarian” presents the results of a questionnaire study among veterinarians in Brazil, investigating their emotional reactions in the context of animal patients’ deaths.
Starting from the idea of veterinarians being affected as secondary victims by their patients’ death, the authors created a 20-item questionnaire that presented questions about the veterinarian participants’ decisions, feelings, and actions when dealing with the loss of a patient. The data for three of the 20 items were analysed statistically and presented in the results part of the article. The authors found gender differences in the participants’ answers as well as differences depending on their time of professional experience, age, specialisation, and the number of animals the veterinarians had already euthanised. Additionally, differences in coping and communication strategies were influential for the emotional affectedness of participants.
The authors discuss the effect of communication training, and look into several reasons for the detected differences between demographic groups.
The studied topic is of high relevance to the veterinary field and the presented study, accordingly, presents valuable insights for the readership. Nevertheless, I would suggest some minor changes.
I would like to learn more about the methodology to fully understand the study. The authors mention a 20-items questionnaire but the do not explain what type of questions (Open-ended? Multiple choice? Likert scale?) it includes. It would be helpful to add the questionnaire to the attachment. Additionally, only three items were selected for the analysis presented in this article. The choice is explained in the methods section. However, without knowing the type and content of the other questions the selection seems a bit arbitrary to me.
The presentation of the results is a bit too detailed, in my opinion, to be able to understand which of the findings are relevant for the analysis and discussion. Maybe presenting a table with the results and just elaborating on the most striking ones would give a better overview. But that is a matter of personal preferences, I guess.
In the discussion, I miss some references to closely related work. First, I would mention Anne Quain (see, e.g., Quain A. The Gift: Ethically Indicated Euthanasia in Companion Animal Practice. Veterinary Sciences. 2021; 8(8):141. https://doi.org/10.3390/vetsci8080141) who discusses the meaning of euthanasia for veterinarians, especially also the role of the vet as an “executioner” which might relate to the guilt/shame the participants in the here presented study seem to feel for the animal’s death. Then, I suggest including the findings from an article dealing with quite similar aspects including gender differences in Austrian veterinarians by Hartnack et al. (Hartnack, S., Springer, S., Pittavino, M. et al. Attitudes of Austrian veterinarians towards euthanasia in small animal practice: impacts of age and gender on views on euthanasia. BMC Vet Res 12, 26 (2016). https://doi.org/10.1186/s12917-016-0649-0). Finally, there actually is literature (even empirical) on the comparison between human and animal patients’ death in human and veterinary medicine, respectively (Selter, Felicitas, et al. "End-of-life decisions: A focus group study with German health professionals from human and veterinary medicine." Frontiers in Veterinary Science 10 (2023): 1044561.). Including these articles might help to discuss a few more potential explanations for the findings presented in the results section.
Detailed comments:
l. 44: Wouldn’t the first victim here be the animal who died?
l. 106: please explain what type of questions you used, here (open-ended, choice of answers…?), see my general comment above.
l. 130: Did the participants have other options for “gender” besides female and male? That is, did you provide further choices (like “divers”, “non-binary” or “other”) but the participants did not choose them, or did you just provide “female” and “male” as options?
l. 163: The reference of “they” is unclear, here. Grammatically, it would be “males” but from the content it should be “females”, shouldn’t it?
l. 172: Should it be “reach” instead of “reaches”?
l. 244: “how they felt” or “how they feel”?
I would like to thank the authors for their research in this highly relevant topic within veterinary ethics!
Author Response
The article „The emotional impact of patient loss on Brazilian veterinarian” presents the results of a questionnaire study among veterinarians in Brazil, investigating their emotional reactions in the context of animal patients’ deaths.
Starting from the idea of veterinarians being affected as secondary victims by their patients’ death, the authors created a 20-item questionnaire that presented questions about the veterinarian participants’ decisions, feelings, and actions when dealing with the loss of a patient. The data for three of the 20 items were analysed statistically and presented in the results part of the article. The authors found gender differences in the participants’ answers as well as differences depending on their time of professional experience, age, specialisation, and the number of animals the veterinarians had already euthanised. Additionally, differences in coping and communication strategies were influential for the emotional affectedness of participants.
The authors discuss the effect of communication training, and look into several reasons for the detected differences between demographic groups.
The studied topic is of high relevance to the veterinary field and the presented study, accordingly, presents valuable insights for the readership.
We wish to thank the reviewer for the positive feedback and the useful comments.
Nevertheless, I would suggest some minor changes.
I would like to learn more about the methodology to fully understand the study. The authors mention a 20-items questionnaire but they do not explain what type of questions (Open-ended? Multiple choice? Likert scale?) it includes. It would be helpful to add the questionnaire to the attachment. Additionally, only three items were selected for the analysis presented in this article. The choice is explained in the methods section. However, without knowing the type and content of the other questions the selection seems a bit arbitrary to me.
Dear Reviewer, thank you for your suggestion. The questionnaire was added as supplementary material (the original one in Portuguese as well as its translation in English) and details about items have been added in the manuscript in the Material and methods section (see lines 112-115).
The presentation of the results is a bit too detailed, in my opinion, to be able to understand which of the findings are relevant for the analysis and discussion. Maybe presenting a table with the results and just elaborating on the most striking ones would give a better overview. But that is a matter of personal preferences, I guess.
The reviewer is right. We completely restructured and rephrased the result section. In particular, we eliminated the sentences related to the non-significant findings to make the reading smoother. However, we included 3 tables on the GLMs results so that also not significant findings are reported.
In the discussion, I miss some references to closely related work. First, I would mention Anne Quain (see, e.g., Quain A. The Gift: Ethically Indicated Euthanasia in Companion Animal Practice. Veterinary Sciences. 2021; 8(8):141. https://doi.org/10.3390/vetsci8080141) who discusses the meaning of euthanasia for veterinarians, especially also the role of the vet as an “executioner” which might relate to the guilt/shame the participants in the here presented study seem to feel for the animal’s death. Then, I suggest including the findings from an article dealing with quite similar aspects including gender differences in Austrian veterinarians by Hartnack et al. (Hartnack, S., Springer, S., Pittavino, M. et al. Attitudes of Austrian veterinarians towards euthanasia in small animal practice: impacts of age and gender on views on euthanasia. BMC Vet Res 12, 26 (2016). https://doi.org/10.1186/s12917-016-0649-0). Finally, there actually is literature (even empirical) on the comparison between human and animal patients’ death in human and veterinary medicine, respectively (Selter, Felicitas, et al. "End-of-life decisions: A focus group study with German health professionals from human and veterinary medicine." Frontiers in Veterinary Science 10 (2023): 1044561.). Including these articles might help to discuss a few more potential explanations for the findings presented in the results section.
Dear Reviewer, thanks for your comments.
We change the sentence and add the references you suggested as follow (see lines 591-593): Members of the veterinary team experience depression and other psychological suffering at significant rates and professionals who are female, younger, and employed in small or mixed practices are more likely to experience stress at work and contemplate suicide.
And at line 551-554: At first appearance, human and (companion animal) veterinary medicine appears to share difficult processes in end-of-life (EOL) decision-making. However, there are significant differences between the therapeutic alternatives offered by the two professions
Detailed comments:
- 44: Wouldn’t the first victim here be the animal who died?
Dear Reviewer, thanks for reaising this issue. However, in the veterinary literature the first victim is the caregiver.
- 106: please explain what type of questions you used, here (open-ended, choice of answers…?), see my general comment above.
Such details have been added at lines 112-115.
- 130: Did the participants have other options for “gender” besides female and male? That is, did you provide further choices (like “divers”, “non-binary” or “other”) but the participants did not choose them, or did you just provide “female” and “male” as options?
Dear reviewer, the question about gender only had male and female as options. This point could have been better addressed by authors when preparing the survey but at least now, having the complete questionnaire available, readers can see how the item was asked and its options.
- 163: The reference of “they” is unclear, here. Grammatically, it would be “males” but from the content it should be “females”, shouldn’t it?
We changed “they” in “females”
- 172: Should it be “reach” instead of “reaches”?
We changed according to your suggestion
- 244: “how they felt” or “how they feel”?
We modified in “how they felt”
I would like to thank the authors for their research in this highly relevant topic within veterinary ethics!
We thank you for your thoughtful suggestions you made to help us in underlying these important points.

Reviewer 3 Report
Comments and Suggestions for Authors
Introduction: Since the research focuses on the veterinarian, there is no need to go details regarding the pet owners. (page 2: lines 50-63). It would be better to focus more specifically on the work that has been done within the target population and provide more detail with regard to the nature of the roles and responsibilities that veterinarians and their staff play (page 2: lines 76-78 and lines 91-92).
Materials and Methods: Please provide details about the 20-item questionnaire that was used to collect the information. Include the questions and the response categories to those questions.
Statistical Analysis: It is not possible to assess the appropriateness of the statistical tests that were used without details about the variables that were collected.
Results: The authors need to provide data in tables. It is difficult to read the numbers and percentages when they are presented in the text alone.
page 3: lines 135-137-categories reported are overlapping (5-10 years; 10-20 years).
page 4: line 153: wanted to do-delete the word do; watering should be tearing up
page 4: lines 167-196-Two paragraphs are repeated.
pages 4-5: lines 156-239-data used in these paragraphs should be presented in tables.
Discussion: The authors make a point to discuss training received in school in relation to communication strategies but have not provided the actual data regarding this in the results section of the manuscript, although they presented statistical test results.
page 10; line 323-324-the authors suggest that the fact that a higher proportion of the sample were female than male was a possible bias, but they did not provide data on the overall percentage of veterinarians in practice in Brazil were female. Therefore, it is not possible to assess whether there this was a biased sample or not.
Comments on the Quality of English Language1. "watering" should be tearing up in the text and table
2. "didn't cry but wanted to do" delete do
3. "Difficult to talk with" did you mean to talk to?
4. "I never had the problem...because I have the control of the situation" I have control of the situation.
5. page 6 line 251: "this topic received little attention"- this topic has received little attention
Author Response
Comments and Suggestions for Authors
Introduction: Since the research focuses on the veterinarian, there is no need to go details regarding the pet owners. (page 2: lines 50-63). It would be better to focus more specifically on the work that has been done within the target population and provide more detail with regard to the nature of the roles and responsibilities that veterinarians and their staff play (page 2: lines 76-78 and lines 91-92).
Lines 50-63: Dear Reviewer, thanks for your suggestion. However, we consider giving some details on the pet owners is relevant, since the emotional impact started also by increased expectations for the highest quality veterinary care. We would like to keep these two sentences but we provide also more details with regard to the nature of the roles and responsibilities that veterinarians and their staff play. We have changed the sentences in lines 80-87 and 101-107.
Lines 80-87: Effective client support during and after a patient's death is characterized by empathic communication. Veterinary care should prioritize improving patient comfort and minimizing suffering while fostering a cooperative and supportive collaboration with the caregiver client [16]. Relationship-centered care, which is defined as a partnership in which the veterinarian and the client use discussion and collaborative decision-making to incorporate the client's perspectives and recognition of the importance of animals in the client's life into all aspects of care, has been suggested as the ideal model for veterinarian-client relationships, just like in human medicine
Lines 101-107: The abilities required for excellent end-of-life treatment have gone beyond what many veterinarian offices are qualified to deliver. Under any circumstances, the healthcare team must communicate with a unified voice and sense of purpose, especially when end-of-life care is involved. In end-of-life situations, each member of the veterinary healthcare team should have clearly defined caregiving and client-support roles, particularly ones that make use of individual abilities, strengths, and experience.
Materials and Methods: Please provide details about the 20-item questionnaire that was used to collect the information. Include the questions and the response categories to those questions.
Dear Reviewer, we added the questionnaire as supplementary material (the original one in Portuguese and translated into English) in order to give to the reader a clearer picture of the methodology used.
Details about the type of items have no been reported in the text at lines 112-115: All questions were close-ended, so respondents were allowed to choose one (all but items 10, 11 and 15) or more answers from a given list of answer options. All items were mandatory; only item 15 had to be answered by respondents who chose “yes” to item 14.
Statistical Analysis: It is not possible to assess the appropriateness of the statistical tests that were used without details about the variables that were collected.
We thank the reviewer for the comment. We revised the statistical analysis section by adding further details on the procedure followed (Line 150-150). Finally, we reported all the results from the GLMS in 3 different tables that we included in the text. We hope we were able to improve the manuscript by following the reviewer’s input.
Results: The authors need to provide data in tables. It is difficult to read the numbers and percentages when they are presented in the text alone.
Tables 1, 3 and 5 have been added as suggested.
page 3: lines 135-137-categories reported are overlapping (5-10 years; 10-20 years).
Dear Reviewer, thank you for spotting this. We revised the sentence reporting the categories as asked in the questionnaire.
page 4: line 153: wanted to do-delete the word do; watering should be tearing up
Thank you, the words have been changed.
page 4: lines 167-196-Two paragraphs are repeated.
Dear Reviewer, Thank you very much for pointing this out. Anyway, the whole paragraph has been dramatically changed so the sentence is not present anymore.
Pages 4-5: lines 156-239-data used in these paragraphs should be presented in tables.
Tables 3 and 5 have been added to present those data.
Discussion: The authors make a point to discuss training received in school in relation to communication strategies but have not provided the actual data regarding this in the results section of the manuscript, although they presented statistical test results.
The actual data are now presented in Table 3.
page 10; line 323-324-the authors suggest that the fact that a higher proportion of the sample were female than male was a possible bias, but they did not provide data on the overall percentage of veterinarians in practice in Brazil were female. Therefore, it is not possible to assess whether there this was a biased sample or not.
Dear Reviewer, the proportion of genders in Brazilian is not known, however we agree with you that having a high number of female vets is not necessarily a bias, so we have modified the text accordingly (see lines 633-634).
Comments on the Quality of English Language
- "watering" should be tearing up in the text and table
- "didn't cry but wanted to do" delete do
- "Difficult to talk with" did you mean to talk to?
- "I never had the problem...because I have the control of the situation" I have control of the situation.
- page 6 line 251: "this topic received little attention"- this topic has received little attention
Dear Reviewer, We apologize for this and thank you for identifying these errors, that have now been addressed.

Reviewer 4 Report
Comments and Suggestions for Authors
The authors have addressed a significant issue with data, not just opinion, and offer valuable suggestions for the improvement of veterinary education and professional behaviour. Points for the authors to consider in a revision include:
Major points
1. Age and time since graduation are two of the study variables. I would expect these to be strongly confounded or linked, with older veterinarians also likely to have longer times since graduation. Perhaps check for a correlation between these variables and, if they are strongly correlated as I suspect, use one rather than both in the analyses. This has implications for the GLM.
2. In the discussion of reactions of the veterinarians to communicating bad news (text, table 1) the percentages for males, females and the totals each sums to well over 100%, indicating that one respondent may have multiple reactions. This should be clarified in the table caption and when introducing the relevant data in the text. It would also be interesting to know how many in the sample experienced multiple reactions, and which reactions were commonly grouped together.
3. I don’t understand what is meant by ‘watering’ (Table 1, also mentioned in the text). My best guess is that the meaning is ‘watering eyes’ or ‘teary,’ meaning on the verge of crying but not crying openly.
4. Sections 3.1, 3.2 and 3.3 in the results include a lot of dense text. I wondered if this text could be better presented in tables or graphs. For example, could the statistical comparisons presented in the text of Section 3.2 be given in matrix tables comparing males and females, or different ages? Such a tablular comparison could also allow you to show clearly where Bonferroni corrections were applied and how. Please also indicate whether you used a sequential Bonferroni correction or a standard one.
5. Perhaps I need new glasses, but I didn’t see a reference to the questionnaire being included in supplementary information. It would be most valuable to researchers planning similar studies to see this. It would be handy to give it in the original Portugese, so that no ambiguity is introduced in translation.
Minor points
1. Authors 1 and 4 do not give an instituitonal afficiliation. If possible, it is helpful to do so. If they are retired or in private practice, that can be stated.
2. Line 39 – replace ‘not uncommon’ with ‘common.’ Double negatives may cause ambiguity.
3. Line 68 – instead of ‘theoretical education’ just use ‘education.’ One could argue that dealing with stresses in the workplace is practical, not theoretical.
4. Line 82 – add a comma after ‘recklessly.’
5. Line 89 – delete ‘that generate great.’ Veterinary workplaces have their share of trauma and suffering, but the sentence as written implies tht the veterinarians cause the suffering.
6. Line 117 – insert ‘and’ as indicated: ‘… they tried to talk to, and if they had …’
7. Line 134 – ‘had graduated’
8. Line 163 – replace ‘However, they’ with ‘However, females’
9. Lines 203-294 – I didn’t understand the value of the sentence ‘Other reported causes were … fractures.’ If that sentence is deleted, the important comment that specialists see more trauma cases and terminal illnesses than specialists is still clear. I assume that reference (30) establishes this point. Please consider deleting the sentence indicated.
Comments on the Quality of English LanguageEnglish language is fine, with only small modifications suggested. If I was as fluent in Portugese or Italian as the authors are in English, I'd be pleased.
Author Response
Comments and Suggestions for Authors
The authors have addressed a significant issue with data, not just opinion, and offer valuable suggestions for the improvement of veterinary education and professional behaviour.
Dear Reviewer, We are very grateful for your detailed and beneficial remarks on our manuscript.
Points for the authors to consider in a revision include:
Major points
- Age and time since graduation are two of the study variables. I would expect these to be strongly confounded or linked, with older veterinarians also likely to have longer times since graduation. Perhaps check for a correlation between these variables and, if they are strongly correlated as I suspect, use one rather than both in the analyses. This has implications for the GLM.
We thank the reviewer for the comment. We performed multicollinearity diagnostics before including the variables in the GLM and we found the highest VIF for age and time from graduation, likely because, as the reviewer correctly assumed, they are correlated to some extent. However, the VIF values were 2.731 and 2.887 that, to our knowledge, are still below the threshold that indicates strong multicollinearity. This is the reason why we decided to include both variables in the model. Nonetheless, we followed the reviewer’s advice and after checking for correlation between age and time from graduation (Kendall’s Tau=>0.7), we ran GLMs without the variable “time from graduation”. In the new model for item 13 the results are almost identical to the previous model. In the new model for item 18 the results are similar with two previous almost significant results that turned into a p value <0.05.
Since the AICc values (item 13: old model=496,875, new model=313,310, item 18: old model= 482,473, new model=317,572) suggest a better fit for the new models in both cases, we will follow the reviewer’s suggestion and keep the models without the “time from graduation” variable.
More detailed information about the statistical procedure was included in the statistical analysis section (line 121-168)
- In the discussion of reactions of the veterinarians to communicating bad news (text, table 1) the percentages for males, females and the totals each sums to well over 100%, indicating that one respondent may have multiple reactions. This should be clarified in the table caption and when introducing the relevant data in the text. It would also be interesting to know how many in the sample experienced multiple reactions, and which reactions were commonly grouped together.
Dear Reviewer as reported in lines 112-115, “All questions were close-ended, so respondents were allowed to choose one (all but items 10, 11 and 15) or more answers from a given list of answer options. All items were mandatory; only item 15 had to be answered by respondents who chose “yes” to item 14.” We also believe that it would be interesting to find out how many people in the sample had more than one reaction, as well as which reactions were most frequently combined. Response frequencies (the number of times a response has been given based on the total response; so, responses may exceed the total number of respondents) have been added to tables 2 and 4. But a follow-up manuscript would most likely develop these.
- I don’t understand what is meant by ‘watering’ (Table 1, also mentioned in the text). My best guess is that the meaning is ‘watering eyes’ or ‘teary,’ meaning on the verge of crying but not crying openly.
Dear Reviewer, We changed the word “watering” with “tearing up”.
- Sections 3.1, 3.2 and 3.3 in the results include a lot of dense text. I wondered if this text could be better presented in tables or graphs. For example, could the statistical comparisons presented in the text of Section 3.2 be given in matrix tables comparing males and females, or different ages? Such a tablular comparison could also allow you to show clearly where Bonferroni corrections were applied and how. Please also indicate whether you used a sequential Bonferroni correction or a standard one.
We thank the reviewer for the suggestion. We restructured the whole results section in order to present the results in a more fluid and understandable way. We also added 3 tables to summarize the results from the GLMs. As for the significance correction standard Bonferroni was applied (not the Holm-Bonferroni method) for all post-hoc tests.
- Perhaps I need new glasses, but I didn’t see a reference to the questionnaire being included in supplementary information. It would be most valuable to researchers planning similar studies to see this. It would be handy to give it in the original Portugese, so that no ambiguity is introduced in translation.
Dear Reviewer, thank you for your suggestion. The original questionnaire in Portoguese, as well as its English version, were included in the supplementary material.
Minor points
- Authors 1 and 4 do not give an instituitonal afficiliation. If possible, it is helpful to do so. If they are retired or in private practice, that can be stated.
Dear Reviewer, thank you for your suggestion. Author 1 and 4 have a private practice and do not have an institutional affiliation.
- Line 39 – replace ‘not uncommon’ with ‘common.’ Double negatives may cause ambiguity.
Thank you very much for pointing this out,it has been modified.
- Line 68 – instead of ‘theoretical education’ just use ‘education.’ One could argue that dealing with stresses in the workplace is practical, not theoretical.
We revised it according to your suggestion.
- Line 82 – add a comma after ‘recklessly.’
We revised it according to your suggestion.
- Line 89 – delete ‘that generate great.’ Veterinary workplaces have their share of trauma and suffering, but the sentence as written implies tht the veterinarians cause the suffering.
We revised it according to your suggestion
- Line 117 – insert ‘and’ as indicated: ‘… they tried to talk to, and if they had …’
We revised it according to your suggestion
- Line 134 – ‘had graduated’
We revised it according to your suggestion
- Line 163 – replace ‘However, they’ with ‘However, females’
We revised it according to your suggestion
- Lines 203-294 – I didn’t understand the value of the sentence ‘Other reported causes were … fractures.’ If that sentence is deleted, the important comment that specialists see more trauma cases and terminal illnesses than specialists is still clear. I assume that reference (30) establishes this point. Please consider deleting the sentence indicated.
The sentence has been deleted
Comments on the Quality of English Language
English language is fine, with only small modifications suggested. If I was as fluent in Portugese or Italian as the authors are in English, I'd be pleased.
We have revised the text also for language.

Round 2
Reviewer 3 Report
Comments and Suggestions for Authors
Thank you for your careful consideration of the review comments.
Reviewer 4 Report
Comments and Suggestions for Authors
Thank you for addressing all comments thoroughly. In particular, the revised presentation of the results is effective. I have no further suggestions.